# Inhibition of Peroxiredoxin 6 PLA2 Activity Decreases Oxidative Stress and the Severity of Acute Lung Injury in the Mouse Cecal Ligation and Puncture Model

**DOI:** 10.3390/antiox10111676

**Published:** 2021-10-24

**Authors:** Aron B. Fisher, Chandra Dodia, Jian-Qin Tao, Sheldon I. Feinstein, Shampa Chatterjee

**Affiliations:** 1Department of Physiology and Institute for Environmental Medicine, University of Pennsylvania Perelman School of Medicine, Philadelphia, PA 19104, USA; cdodia@pennmedicine.upenn.edu (C.D.); taoj@pennmedicine.upenn.edu (J.-Q.T.); shampac@pennmedicine.upenn.edu (S.C.); 2Peroxitech, Inc., Philadelphia, PA 19104, USA; sifinphilly@gmail.com

**Keywords:** acute lung injury, peroxiredoxin 6, phospholipase A2, NADPH oxidase, oxidative lung injury, rac1 and 2, bacterial pneumonia, reactive oxygen species

## Abstract

The use of agents to inhibit the production of reactive oxygen species (ROS) has been proposed for the treatment of Acute Lung Injury (ALI). However, this approach also inhibits the bactericidal activity of polymorphonuclear leucocytes (PMN) and other cells, raising the possibility of aggravating lung injury in ALI associated with bacterial infection. We used the cecal ligation and puncture (CLP) model of ALI associated with sepsis to investigate the effect of inhibiting NADPH oxidase 2 (NOX2)-derived ROS production, the main source of ROS in lungs. A phospholipase A_2_ inhibitor called peroxiredoxin 6 inhibitory peptide-2 (PIP-2) was used to inhibit NOX2 activation; the peptide prevents liberation of Rac, a necessary NOX2 co-factor. At 18 h after intravenous treatment with 2 µg PIP-2 /gram body weight (wt), the number of colony-forming bacteria in lungs and peritoneal fluid of mice with CLP was approximately doubled as compared to untreated mice. Treatment with 10 µg PIP-2/g body wt resulted in 100% mortality within 18 h. Antibiotic treatment abolished both the increase in lung bacteria with low dose PIP-2 and the increased mortality with high dose PIP-2. Treatment with PIP-2 plus antibiotics resulted in significantly improved lung histology, decreased PMN infiltration, decreased lung fluid accumulation, and decreased oxidative lung injury compared to antibiotics alone. We conclude that the administration of PIP-2 provides partial protection against lung injury in a model of ALI due to bacterial infection, while concurrent antibiotic treatment abolishes the deleterious effects of PIP-2 on lung bacterial clearance. These results suggest that addition of PIP-2 to the antibiotic regimen is beneficial for treatment of ALI associated with bacterial infection.

## 1. Introduction

The Acute Lung Injury (ALI) syndrome occurs through complex mechanisms from diverse etiologies [1,2]. Although the pathogenesis of ALI is multifactorial, the role of inflammation with the production of reactive oxygen species (ROS) is an important component within essentially all etiologies [3,4,5,6,7,8,9]. ROS serve to signal for initiation of the inflammatory cascade, and they also exert toxicity through the oxidation of tissue macromolecules (lipids, proteins, DNA). The major source of ROS in the lung is NADPH oxidase type 2 (NOX2) that is expressed especially in endothelial cells, polymorphonuclear leucocytes (PMN) and alveolar macrophages, but also in lung epithelium [4,7,9,10,11]. This normally quiescent enzyme requires activation via a complex pathway that includes the binding of cytoplasmic factors p40, p47, p67, and Rac in order to generate ROS [10,12,13,14,15,16]. A variety of agents have been shown to inhibit the activation of NOX2 and to modulate the course of lung injury [17,18,19,20,21,22,23,24]. NOX2 inhibition has also been effective in limiting injury in other organs. For example, the inhibition of NOX2 activity (with apocynin) was protective in a mouse model of lipopolysaccharide (LPS)-induced cardiomyopathy [25]. Importantly, the inhibition of ROS production is a far more effective way to prevent oxidant stress as compared to treatment with antioxidants, since the latter are effectively competing with normal tissue components for interaction with oxidants; thus, oxidation of tissue components (oxidant stress) is the inevitable result of tissue-mediated ROS scavenging. However, currently available agents to inhibit NOX2 activity are either potentially toxic or otherwise not suitable for development as therapeutic drugs, and no agent is currently FDA approved for primary treatment of ALI [26,27].

We have shown that the activation of NOX2 in mouse lungs, alveolar macrophages, and polymorphonuclear leukocytes (PMN) requires the phospholipase A_2_ activity of peroxiredoxin 6 (Prdx6), and that the inhibition of this activity (called aiPLA_2_ activity) largely prevents ROS generation by lung cells [28,29]. This role of aiPLA_2_ was confirmed by genetic inactivation of aiPLA_2_ that prevented NOX2 activation [30]. The aiPLA_2_ activation process is initiated through the phosphorylation of cytosolic Prdx6 and its translocation to the cell membrane, where its PLA_2_ activity results in the generation of lysophosphatidylcholine and, subsequently, the release of Rac [28,29,31], one of the cytoplasmic factors that are required for NOX2 activation, as described above. Note that Rac2 is the required co-factor in PMN, while Rac1 is required in endothelial and other lung cells, but release of either form of Rac appears to require aiPLA_2_ activity and is inhibited by PIP-2 [32].

PLA_2_ represents a large family of enzymes that exhibit varying cellular and substrate specificities [33,34,35]. Previous studies have shown that one or more of these PLA_2_ enzymes can lead to NOX2 activation through the liberation of arachidonic acid; however, the specific enzymes involved and the physiological roles of activation by these other PLA_2_ enzymes remain unclear [36,37]. To inhibit aiPLA_2_ activity pharmacologically, we initially used a non-specific PLA_2_ inhibitor, called MJ33, and found protection against lung injury in mice with LPS-induced ALI or with exposure to hyperoxia [38,39]. Inhibition of aiPLA_2_ activity with MJ33 also was shown by others to protect against inflammatory brain injury in a rat model of acute stroke [40]. Our studies further showed that surfactant protein A (SP-A), a naturally occurring lung protein, binds to Prdx6 and inhibits its aiPLA_2_ activity [41,42]. Based on this observation, we developed a peptide inhibitor of aiPLA_2_. We initially identified a 16 amino acid (aa) peptide derived from SP-A and published its binding kinetics with Prdx6 [43]. We subsequently described a 9 aa peptide within the 16 aa sequence as the minimal sequence for inhibition of aiPLA_2_ activity [44]. We have called this peptide peroxiredoxin 6-inhibitory peptide-2 or PIP-2 (PIP-1 is the analogous peptide derived from mouse/rat lung surfactant protein) [44]. PIP-2 appears to be specific for the inhibition of aiPLA_2_ activity and its interaction with other PLA_2_ enzymes is unlikely. We have shown that PIP-2 protects against lung injury in both LPS and ventilator-induced lung injury (VILI) models of ALI in mice [32,45].

While the results with the LPS and VILI models showed the ability of PIP-2 to prevent or treat ALI in mice, it is important to note that these are sterile models of lung injury. Bacterial infection of the lung (pneumonia) is a major cause of ALI in a significant fraction of human cases [46]. Since NOX2 and its generation of ROS (i.e., superoxide anion) plays a significant role in bacterial killing by PMN, macrophages, and possibly other cells, treatment with PIP-2 could be counter-productive in ALI associated with bacterial infection by interfering with cellular bactericidal activity [25]. The purposes of the present study were: 1) to evaluate the effect of PIP-2 itself on bacterial clearance and, 2) to determine the effect of PIP-2 +/− antibiotics in the treatment of ALI associated with bacterial infection.

## 2. Materials and Methods

### 2.1. Mice

Male C57Bl/6 mice were obtained at approximately 8 weeks of age from Jackson Laboratories (Bar Harbor, ME, USA) and maintained in our vivarium for 1–2 weeks on a diet of standard chow and under high-efficiency particulate filtered air (HEPA) at a temperature of 25 ± 3 °C. For experiments, mice were anesthetized with ketamine (100 µg/g body wt), xylazine (4 µg/g) and acepromazine (1 µg/g) injected intraperitoneally (IP). In some experiments, an antibiotic ‘cocktail’ of imepenem (25 µg/g body wt), gentamycin (50 µg/g), and vancomycin (50 µg/g) was given IP. PIP-2, when administered, was given by IV bolus injection using the retinal vein. All procedures related to the study of mice were approved by the University of Pennsylvania Animal Care and Use Committee (IACUC).

### 2.2. PIP-2

PIP-2 was synthesized by APeptide Corp. (Shanghai, China) as the trifluoroacetate salt at >90% purity as shown by both mass spectrometry and HPLC results provided by the manufacturer. The amino acid sequence of PIP-2, LHDFRHQIL, has been published previously [32,44]. The peptide was stored at −20 °C although we have shown long term stability (> 10 months) for storage at room temperature. For administration to mice, the peptide was encapsulated in liposomes [dipalmitoyl phosphatidylcholine (DPPC): egg PC: egg phosphatidylglycerol (PG): cholesterol (0.5:0.25:0.10:0.15, mol fraction] as previously described [45]; we have called these lung surfactant-like liposomes. For encapsulation, PIP-2 was added to saline for low dose liposomes and at 5 times the concentration to prepare the high dose liposomes. The encapsulation efficiency was not measured in these experiments. The liposomes were suspended at 14 mg lipid per ml of phosphate-buffered saline (PBS), equivalent to 2 (low dose) or 10 (high dose) mg PIP-2 per ml of PBS. Liposomes were stored overnight at 4 °C, and used within 24 h of preparation. The 2 or 10 mg PIP-2 per ml saline concentration represents both liposome encapsulated and non-encapsulated PIP-2. We have found that only liposome-encapsulated PIP-2 is taken up by lungs [32] so the actual dosage delivered to the lung in a form for possible cellular uptake is lower than the nominal dosage that was administered.

### 2.3. Generation of CLP and Experimental Plan

CLP has been widely used as a model for the study of sepsis-induced acute lung injury [47]. The generation of CLP was based on previous descriptions of the procedure [47]. The abdomen was incised, and the cecum was identified, ligated and punctured 6 times with a 26-gauge (26G) needle. The cecal contents were gently extruded and the abdominal incision was closed. Mice were given 1 mL of warm sterile normal saline subcutaneously for hydration. Mice were treated with one of 6 protocols: (a) CLP alone; (b) & (c) CLP + PIP-2 at either low (2 µg/g body weight) or high (10 µg/g body weight) dose by IV bolus injection.; (d) CLP + antibiotics; or (e) & (f) CLP + antibiotics + PIP-2 at low or high dose. We used two different cohorts of CLP mice for experiments. For cohort 1, PIP-2 was administered just before surgery and antibiotics were administered at the end of surgery after closure of the abdomen. Mice were sacrificed by an overdose of anesthetic at 6 h after surgery. For cohort 2, antibiotic and/or PIP-2 treatment was given at 6 h after surgery and mice were sacrificed ~18 h later (after 24 h of CLP).

After sacrifice of the mice, the thorax and abdomen were incised, and peritoneal fluid was obtained from the peritoneal cavity using a syringe. The lungs were removed from the thorax and ½ of each of the two lungs were combined and homogenized under N_2_ in 0.5 mL phosphate buffered saline (PBS); these homogenates were used for bacterial culture. A small tissue sample (~0.25 g) was removed from the lower lobe of the left lung, for measurement of wet to dry weight ratio (*W/D*). The remaining lung tissue was homogenized under N_2_ and was stored at −80 °C for subsequent biochemical assays. In a small number of experiments, mice were injected IV with 2 or 10µg PIP-2/g body weight, were sacrificed 2 or 24 h later, and lungs, liver, kidney and heart were isolated for subsequent measurement of aiPLA_2_ activity in the organ homogenate.

### 2.4. Bacterial Culture

Undiluted samples and serial dilutions of the lung homogenate and peritoneal fluid were spread on MacConkey agar plates. After overnight incubation at 37 °C, the colonies were counted and the number of viable bacterial cells in the lung homogenate or peritoneal fluid was calculated.

### 2.5. Biochemical Assays

The assay for aiPLA_2_ activity measured the release of ^3^H from lung surfactant-like liposomes with the ^3^H label in the 9–10 position of the *sn*-2 palmitate of DPPC; the assay was carried out in acidic (pH 4), Ca^2+^-free buffer as described previously [29,48]. To measure lung wet to dry ratio (*W/D*), the tissue sample was lightly blotted, weighed (wet weight), and then dried to constant weight (dry weight) in an oven at 60 °C (about 3–4 days); the values obtained are slightly lower than our recent reports because of the addition of the lung blotting step [32]. The frozen lung homogenate was thawed and used for assay of markers of lung inflammation and injury. Lung myeloperoxidase (MPO) activity as an index for PMN accumulation was measured in the lung homogenate by ELISA assay using a commercial kit (Invitrogen, ThermoFisher Scientific, Frederic, MD, USA). The cytokine (MIP-2, IL-6, and TNFα) content in the lung homogenate also was measured by ELISA assay using a commercial assay kit (Oxidative Stress Elisa Strip Profiling Assay, Signosis, Santa Clara, CA, USA) as described previously [32]. To determine oxidative stress, thiobarbituric acid reactive substances (TBARS), 8-isoprostanes, and protein carbonyls were measured in the lung homogenate as described previously [38,39,49].

### 2.6. Lung Histology

A separate group of mice were subjected to CLP using the protocol for cohort 2 with low dose PIP-2 at 6 h and sacrifice at 24 h for evaluation of lung histology. Lungs were excised and fixed in 4% paraformaldehyde. Fixed tissue was sent for processing to the Histology Core of the Children’s Hospital of Philadelphia where the tissue samples were washed with ethanol and then embedded in paraffin. Paraffin blocks were sectioned using a microtome and sections were stained with hematoxylin and eosin (H&E). For each mouse lung, 3–4 sections were obtained, and 10 fields were examined per section. Digital images were acquired with an Aperio image Scope (Leica Biosystems, Chicago, IL, USA) and were visualized at magnifications of 1× and 20× [50,51]. Injury scores were given based on image assessment by computationally deriving 10 sections that were assessed at 60×. The parameters for injury assessment were: **a**. cellular infiltration on a scale of 0–2 with 0 = no infiltrate, 1 = infiltrate in the perivascular compartment, and 2 = infiltrate in perivascular and alveolar compartments); **b**. mean linear intercept (MLI) to indicate disruption of the alveolar-capillary membrane; and **c**. proteinaceous alveolar exudate, using a scale of 0–5 (as a % of the area involved; 0 = no exudate; 5 = exudate in 80% of alveolar area).

### 2.7. Statistical Analysis

Results are presented as mean *±* standard deviation (SD). Group differences were evaluated by 2-tailed t-test or one-way ANOVA followed by a post hoc *t*-test with Bonferroni correction as appropriate. Statistical significance for all studies was accepted as *p* < 0.05.

## 3. Results

### 3.1. Tissue Uptake of PIP-2

The tissue uptake of PIP-2 in liposomes was determined by its inhibition of endogenous aiPLA_2_ activity in the homogenate of lung, heart, liver, and kidney. Under control conditions, lungs had higher expression levels of aiPLA_2_ than the other three organs, with the kidney at ~ 81%, liver at ~ 65%, and heart at ~ 59% compared to lung (Table 1). In all four organs, PIP-2 at the nominal 2 µg/g mouse body weight dose inhibited aiPLA_2_ activity by ~ 80% at 2 h after IV administration. A 10-fold increase in the dose of PIP-2 had no greater effect on aiPLA_2_ activity at 2 h after dosing in any of the four organs, indicating that 2 µg/g PIP-2 IV gave maximal inhibition. Inhibition of aiPLA_2_ activity at 24 h after low dose PIP-2 was similar to the 2 h value for all four organs (Table 1). Thus, there was no decay in the inhibition of intracellular aiPLA_2_ activity during the initial 24 h after IV administration of PIP-2. As pointed out above, the dose used of PIP-2 (2 µg/g) is a nominal dose since encapsulation efficiency was not measured. The present data for the lung confirms our previous results showing no changes in the inhibition of lung aiPLA2 activity between 2 and 24 h; in the previous study, the ½ time for inhibition of lung aiPLA2 activity after IV PIP-2 was ~54 h [44].

Our previous studies showed that angiotensin II-stimulated ROS production by control lungs when maximally inhibited by MJ33 was similar to the ROS production by lungs from NOX2 null mice [45]. Thus, assuming that there was no effect of MJ33 on other PLA_2_ enzymes in the lung, the present result is compatible with total inhibition of NOX2-mediated lung ROS production by inhibition of aiPLA_2_. The enzyme(s) responsible for the relatively low level of ROS production that was not inhibited by aiPLA_2_ activity was not identified.

### 3.2. Mortality

We used the CLP model to evaluate the effect of PIP-2 administration on the relationship between bacterial infection and mouse mortality. Mice with CLP were treated with zero, low dose (2 µg/g body weight), or high dose (10 µg/g) PIP-2; these 3 groups were studied with or without concurrent antibiotic treatment, making 6 groups in all. The studies shown in Table 2 are for the cohort 2 protocol; no mice in the cohort 1 protocol died within 6 h after surgery. The high dose of PIP-2 was studied even though it had no greater effect on organ aiPLA_2_ activity than the low dose (Table 1) to evaluate whether other pathways possibly might be involved in the response to PIP-2.

The high dose of 10 µg/g PIP-2 is 1/2 of the dose that was used for the dose-effect study shown in Table 1. Mortality within 24 h after CLP surgery (18 h after dosing with antibiotics and/or PIP-2) was < 10% for both the CLP and the low dose PIP-2 +/∓ antibiotics groups (Table 2) but was 100% for the high dose PIP-2 group. Concurrent administration of antibiotics abolished the mortality associated with high dose PIP-2 indicating that the mouse deaths were associated with bacterial infection.

### 3.3. Bacterial Culture

Bacterial colonization of lungs was determined for six conditions with both the cohort 1 and 2 protocols. These conditions are control lungs and five of the six experimental conditions described above; for obvious reason, bacterial cultures were not done for the high dose PIP-2 lungs as all of these mice died during the 24 h post-CLP period. Since mice were not continuously observed, we do not know the exact time of death. Further studies with this protocol were not performed because of humane considerations. Culture of homogenized lungs of the CLP mice at 6 h after surgery (cohort 1) showed a relatively small number of bacterial colonies in the peritoneal fluid and even fewer colonies in the lung (Table 3). There was a modest (< 50%) increase in colony counts in both the peritoneal fluid and lungs of mice with CLP that were treated with PIP-2. However, at 18 h (24 h after surgery, cohort 2), there were significant increases compared with control mice in the bacterial colony count in the peritoneal fluid and in the lung; the number of colonies in both peritoneal fluid and lungs increased about 100% in mice with CLP that were treated with PIP-2 alone (Table 3). The increases in peritoneal fluid and lung bacteria colony counts with CLP were reduced to control levels with antibiotic treatment in both PIP-2 treated and untreated mice (Table 3). Thus, treatment with a NOX2 inhibitor via IV infusion of PIP-2, inhibited clearance of bacteria from the lung, but the resultant increase in bacterial colony count was fully reversed by the administration of antibiotics.

### 3.4. Lung Histology

Four groups of lungs (*n* = 3 per group) from cohort 2 corresponding to CLP alone, CLP + antibiotics, CLP + PIP-2, and CLP + antibiotics + PIP-2 were treated at 6 h and sacrificed at 24 h, following surgery for the study of histologic changes in the lungs. Representative sections show areas of alveolar collapse with cellular infiltrates and alveolar exudates with CLP lungs as compared to control lungs (Figure 1A). There was no apparent change with PIP-2 treatment and a modest improvement with antibiotic treatment. However, treatment with antibiotics plus PIP-2 restored lung histology to nearly normal. The histologic changes were quantitated as described in Methods. CLP resulted in significant increases in lung mean linear intercept (MLI) indicating disruption of alveolar-capillary membranes and in the presence of alveolar exudates (Figure 1B). There also was an increase in cellular infiltrate, presumably reflecting inflammatory cells, although the cell types were not identified. Treatment with PIP-2 alone resulted in a slight improvement in the MLI but also slight increases in cellular infiltrate and alveolar exudates. Treatment with the antibiotic cocktail resulted in a significant improvement in the MLI and a decrease in alveolar exudate and cellular infiltrate, although lungs had residual cellular infiltrate and membrane disruption remained abnormal. Treatment with antibiotics plus PIP-2 resulted in nearly normal lung histology (Figure 1A,B)

### 3.5. Lung MPO Activity and W/D

Lung myeloperoxidase (MPO) levels were used as an index of lung infiltration with PMN following CLP. As compared to control, CLP led to a 3-fold increase in MPO at 6 h and 14-fold increase at 24 h (Table 4). PIP-2 markedly decreased lung MPO at both 6 and 24 h. An even greater decrease compared to PIP-2 was seen with antibiotic treatment alone. Treatment with antibiotics plus PIP-2 reduced lung MPO to values below control. Low dose PIP-2 with antibiotics produced maximal effect.

The lung *W/D* reflects the fluid content of the lung. Lung *W/D* increased significantly in the CLP model compared to the control, indicating lung fluid accumulation during the 24 h post-surgical period (Table 4). PIP-2 treatment (low dose) resulted in a significant decrease in the *W/D* at 24 h compared with CLP alone; there was no further decrease in *W/D* with administration of high dose PIP-2 (plus antibiotics). Treatment with antibiotics, either alone or with PIP-2, had no effect on *W/D* (Table 4).

### 3.6. Lung Cytokines

Selected cytokines (TNF-α, IL-6, and MIP-2) were measured in the lung homogenate at 6 and 24 h for PIP-2 +/− antibiotics that were administered at zero time (Figure 2A) or at 6 h after surgery (Figure 2B), respectively. There were no changes from control in cytokine levels measured at 6 h after CLP (cohort 2, Fig 2A). There was a significant increase in all 3 cytokines measured at 24 h after CLP (cohort 2), that was further increased in the PIP-2-treated (low dose) lungs (Figure 2B) consistent with the increase in bacterial colonies. Treatment with antibiotics abolished the increase in cytokines either in the absence or presence of PIP-2. High dose PIP-2 was not studied in the absence of antibiotics.

### 3.7. Lung Oxidative Stress

We measured lung TBARS and 8-isoprostanes as indices of tissue lipid peroxidation, and protein carbonyls as an index of tissue protein oxidation (Table 5). There were no changes in the three indices of oxidative stress at 6 h after CLP. However, there was a 2.7−3.5-fold increase in these indices at 24 h after CLP, indicating lipid peroxidation and protein oxidation of lung tissue during the 6 to 24 h interval after CLP surgery. PIP-2 at low dose resulted in a marked reduction in the oxidative stress in markers; there was no further change with high dose PIP-2. Antibiotics had essentially no effect on these parameters of oxidative stress.

## 4. Discussion

In our previously reported studies to evaluate the efficacy of PIP-2 using mice, ALI was induced with intratracheal LPS while a sepsis model was generated with intraperitoneal (IP) LPS [44]. PIP-2 also was tested in a mouse model of VILI [32]. Although PIP-2 was markedly beneficial in all three models, they represent “sterile”, i.e., non-infectious, forms of ALI. CLP, on the other hand, results in lung injury in the presence of bacterial infection. As pointed out in a recent study that evaluated the treatment of CLP with an analogue of glucagon-like peptide-1, “it can be assumed that without antibiotic co-treatment, there is no survival benefit…” by treatment with their drug [52]. However, the effect of antibiotic treatment and the effect of their drug on bacterial clearance were not evaluated in that study. The overall postulate for the present study was that PIP-2, through its effect to prevent NOX2 activation and subsequent ROS production, would inhibit the bactericidal activity of PMN and thereby limit clearance of bacteria from the lung. We further postulated that this effect of PIP-2 could be overcome by the administration of appropriate antibiotics. As supporting evidence for the effect of antibiotics on clearance of bacteria in the absence of NOX2, patients with chronic granulomatous disease (CGD) have greatly impaired bactericidal activity due to lack of NOX2 (although this is not ALI) but the lifespan for afflicted individuals has increased significantly since the advent of antibiotics [52]. However, can antibiotics overcome the loss of NOX2-derived ROS under conditions of acute bacterial infection?

The CLP model is theoretically similar to the IP LPS model with the added feature of bacterial infection. As a first step to determine the effect of PIP-2 on lung bacterial clearance, we evaluated the mortality of mice with the CLP model. PIP-2 and/or antibiotics were administered at zero time or at 6 h after surgical generation of CLP. Nearly all mice with CLP that were untreated survived for 24 h (Table 2). CLP mice were treated with PIP-2, with antibiotics alone, or with antibiotics plus PIP-2. We utilized a nominal PIP-2 dose of either 2 µg/g body weight, a dose that was effective in treatment of ALI in the mouse LPS model, or a dose 5 times greater (10 µg/g); the latter was closer to the dose (20 µg/g) that resulted in maximal mouse survival with the IP LPS-induced sepsis model [45]. Treatment of these CLP mice with low dose PIP-2 (2 µg/g) had no effect on mortality, although bacterial culture showed a doubling in the number of bacterial colonies in both the lungs and peritoneal fluid (Table 3). The increase in lung bacterial colony count in PIP-2-treated mice was reversed in the presence of antibiotics (Table 3). These results are consistent with the requirement for ROS generation by NOX2 for bacterial killing. Thus, PIP-2 inhibits the bactericidal efficacy of PMN as well as that of macrophages and possibly other cells associated with immune function resulting, by definition, in increased lung infection. Of note, NOX2-mediated ROS generation is responsible for a considerable fraction of the ROS produced in the lung and accounts for most bactericidal activity [53]. In the absence of antibiotics, the increased bacterial infection with high dose PIP-2 was lethal; however, lethality with the high dose of PIP-2 was prevented by the co-administration of antibiotics.

Although the lethality of high dose PIP-2 in the absence of antibiotics clearly was related to bacterial infection and was prevented by antibiotic treatment, the precise mechanism(s) is unclear. It was not related to a greater degree of NOX2 inhibition in the lung, since low dose PIP-2 was equally effective compared to the high dose as an inhibitor of NOX2, but the low dose of PIP-2 was not lethal. Lethality also was not related to differential effects on other major organs since % and duration of inhibition, at least in the three non-lung organs tested, was similar to the lung. A third possibility is that mortality with high PIP-2 was related to the inhibition of NOX2 in proliferating the PMN that occurred after the low dose of PIP-2 had been cleared from the bloodstream. Thus, at low dose PIP-2, a significant fraction of PMN might have escaped the effects of the inhibitor and could exert at least some bactericidal activity, while a high dose of PIP-2 could have inhibited these later recruited PMN and their NOX2 production; the net result would be a greater inhibition of PMN bactericidal activity with high dose PIP-2. This possibility also seems unlikely since a similar volume of liposomes was injected with the low and high doses of PIP-2 so that the fractional uptake of PIP-2 should have been similar under the 2 conditions. Finally, low dose PIP-2 resulted in increased cytokine release and a possibly increased cytokine release with high dose PIP-2 could have influenced mortality; however, cytokines with high dose PIP-2 alone were not evaluated. Thus, one or more of these possibilities may have been associated with increased mortality, but the actual mechanism for increased mortality with high dose PIP-2 is unclear.

Treatment with antibiotics in both the low and high dose PIP-2 models overcame the loss of NOX2-related bactericidal mechanisms, essentially eliminated the increased bacterial colony counts associated with PIP-2 and prevented sepsis-induced death. Therefore, antibiotics are clearly useful to prevent or treat the bacterial infection but may not prevent or treat the secondary inflammatory changes that characterize the ALI syndrome. We postulate that treatment with PIP-2 is beneficial in preventing lung injury associated with lung inflammation as demonstrated in the present study with CLP and in our previous studies with the LPS and VILI models of ALI [32,45]. In the present experiments, we used two protocols to study the lung effects of PIP-2 treatment in mice with CLP. The first protocol using cohort 1 was to evaluate the effect of PIP-2 during the initial 6 h after the surgery to produce CLP; this represents an ALI preventative strategy. For the second protocol using cohort 2, treatment with PIP-2 began at 6 h after surgery and mice were sacrificed at 24 h; this corresponds to an ALI treatment strategy. The effects of PIP-2 on the manifestations of CLP were determined by evaluation of lung histology, lung edema (lung *W/D*), lung inflammation (MPO, cytokines), and lung oxidative stress (lung TBARS, 8-isoprostanes, and conjugated dienes).

Lung injury at 6 h after CLP was relatively slight and was effectively treated with antibiotics. Thus, a beneficial effect of PIP-2 was not seen at 6 h (the prevention protocol) in these minimally injured lungs. However, studies at 24 h after CLP (the treatment protocol) showed that PIP-2 had a significant beneficial effect on the measured indices of lung injury compared to treatment with antibiotics alone. With PIP-2, there was: (1) improvement in histology (from MLI of 16.9 with antibiotics to 22.4 with addition of PIP-2, control 24.3); (2) decreased PMN infiltration indicated by both infiltration index with histology (antibiotics 8.85 vs. 2.4 with added PIP-2, control 4.90) and MPO measurement (8.94 with antibiotics and 2.40 with the addition of PIP-2, control 4.94); (3) decreased lung edema indicated by *W/D* (7.52 with antibiotics vs. 6.27 with PIP-2 addition, control 4.90); (4) decreased oxidant stress (for antibiotics alone vs. antibiotics plus PIP-2,TBARS 231 vs. 108; isoprostanes 111 vs. 53.4; protein carbonyls,15.8 vs. 7.7); control values for these 3 parameters were 69, 32 and 5.2, respectively. In summary, PIP-2 addition to treatment with antibiotics had a beneficial effect on alveolar disruption and infiltration with PMN, lung fluid accumulation, and tissue oxidation in this CLP model.

Antibiotic treatment alone was effective to reverse the increased lung expression of cytokines in this model of ALI, while PIP-2 by itself led to a modest increase in cytokines. However, this result is different from our previous study with VILI, a non-infectious model of lung injury, that showed a marked inhibition of cytokine expression in the lung lavage fluid by treatment with PIP-2 alone [32]. Moreover, in a recent study with the mouse LPS model, treatment with an anti-inflammatory agent (a GLP-1 analog) reversed the increase in cytokines [54]. The increase in cytokines with PIP-2 treatment alone is presumably related to the increased bacterial load under those conditions, although the precise mechanism is unclear. It should be noted that the antibiotic imipenem, and possibly other antibiotics, when administered long term have some immunomodulatory effect [55]; so, the decreased cytokine release with antibiotics in the present study may possibly be related in part to the immune activity of imipenem.

In contrast to the present results, a previous study of CLP with mice showed that the ‘knockout’ of Prdx6 resulted in significantly greater lung injury as compared with control mice [56]. There are several important differences between the previous study with genetic deletion of Prdx6 and our study with pharmacologic inhibition of Prdx6 PLA_2_ activity. Importantly, Prdx6 expresses a peroxidase activity that plays an important role in antioxidant defense as a major enzyme to reduce phospholipid hydroperoxides and thereby protect cell membranes against oxidative injury [57]. PIP-2 has no effect on this Prdx6 peroxidase activity as the active sites for the two activities (aiPLA_2_ and peroxidase) are totally separate [58]. Therefore, the peroxidase activity is fully active as an antioxidant enzyme in mice treated with PIP-2. In contrast, the peroxidase activity, along with aiPLA_2_ activity, are both deleted with Prdx6 ‘knockout’ so that the antioxidative activity of Prdx6 was lost in the Prdx6 null mice used in the previous study of CLP [56]. Moreover, this previous study did not use antibiotics [56] so, as shown in the present study, increased bacterial infection associated with the absence of aiPLA_2_ would have potentiated the degree of injury.

Thus, the present results indicate that the administration of PIP-2 to block NOX2-mediated ROS production is beneficial for the treatment of patients with ALI associated with bacterial infection. This result is consistent with a previous study showing that the NADPH oxidase inhibitor apocynin attenuated sepsis-induced lung injury in guinea pigs [17]. The benefit to NOX2 inhibition in the present study was present provided that there was adequate antibiotic coverage to prevent the negative effect of PIP-2 on bacterial clearance. Of course, treatment with antibiotics is routine in cases of ALI where bacterial infection (e.g., pneumonia) is suspected. Since ROS and PMN have no direct role in viral clearance, the inhibition of NOX2 should not play a role in viral infections, e.g., influenza, and PIP-2 treatment could be effective in altering the course of lung injury associated with viral disease. Indeed, in a study of mice infected with influenza virus, antioxidant treatment alone, either with a catalase/superoxide mimetic or with a NOX2 inhibitor, was beneficial against lung inflammation and cell death [59]. Another study in mice provided additional support for the use of an antioxidant by demonstrating that the administration of antibiotics plus an inhibitor of NOX2 was more effective than either agent alone in treating influenza with a superimposed bacterial pneumonia [60]. Based on the present results, treatment with PIP-2 should be restricted to low dose if antibiotic resistant bacteria are responsible for the infection.

The results of the present study confirm that PIP-2 inhibits bacterial killing and results in an increased bacterial load in mice with CLP. Although this study used PIP-2 to inhibit NOX2 activation, it is important to note that the adverse effect of PIP-2 on bacterial killing is not specific for PIP-2, but rather is a function of NOX2 inhibition. Thus, any NOX2 inhibitor should have essentially the same effect on bacterial load as shown for PIP-2, although this remains to be tested. Likewise, any effective ROS scavenger will decrease the ROS available for bacterial killing and might decrease the clearance of bacteria. In the present study, the effect of PIP-2 on lung (and peritoneal fluid) bacterial load was overcome fully by treatment with appropriate antibiotics and the combination of PIP-2 plus antibiotics gave significantly greater inhibition of lung injury with the CLP model than the use of antibiotics alone. Thus, the present study confirms that PIP-2 in the presence of adequate antibiotic coverage does not adversely affect bacterial infection of the lung in the CLP model, is not detrimental to bacterial clearance, and helps to protect the lung against oxidative stress and lung injury.

## 5. Conclusions

We show in a mouse model of ALI associated with bacterial infection that treatment with PIP-2 to inhibit NOX2 activation significantly ameliorates lung injury and does not interfere with bacterial clearance in the presence of effective antibiotic coverage. We propose that PIP-2, along with antibiotic therapy, if indicated, could be administered either as treatment for patients with ALI or as prophylaxis to prevent ALI in patients at risk for the syndrome. With current treatment practice, mortality of ALI remains at about 30–40% where it has been for several decades. We propose that the addition of PIP-2 to an effective antibiotic regimen has the potential to inhibit ROS-mediated lung injury and to lower the currently unacceptably high mortality rate for ALI.

## 6. Patent Pending

The University of Pennsylvania has filed a patent application for the 9 amino acid peptide called PIP-2.

## Figures and Tables

**Figure 1 antioxidants-10-01676-f001:**
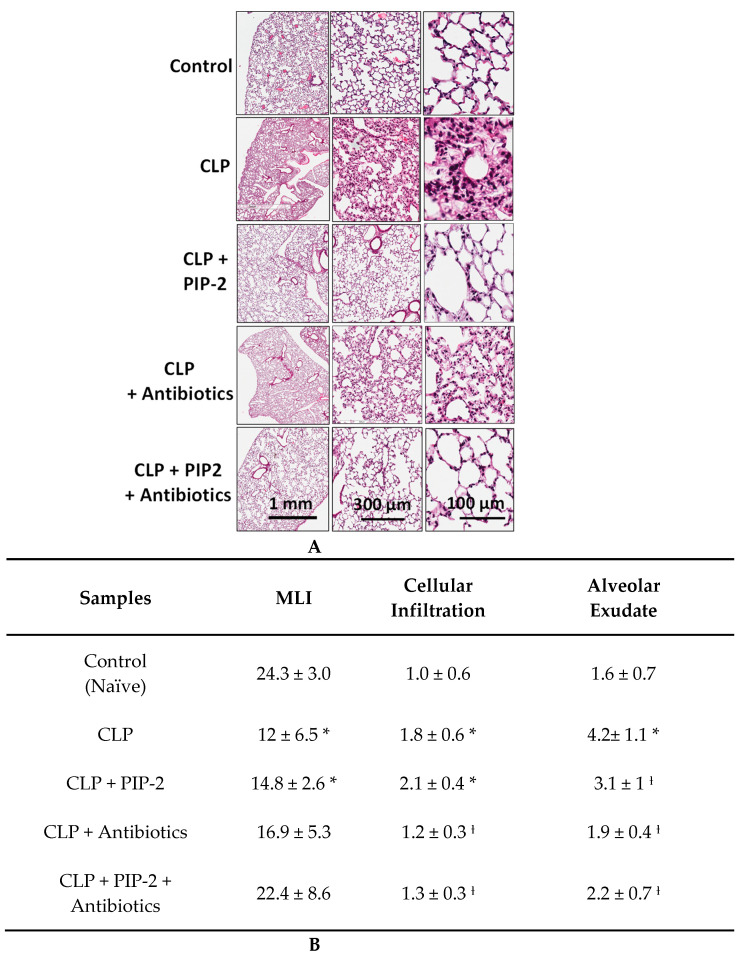
Lung histology with CLP. Mice were treated with antibiotics and/or low dose PIP-2 (2 ug/g) at 6 h after CLP surgery and sacrificed at 24 h. (**A)**. Hematoxylin and eosin (H&E) stained lung sections of mice from each group. Representative images shown are at 1×, 5× and 40× magnification. (**B**). Average injury scores for the different groups. Results are the mean ± SD for *n* = 3 independent experiments (mice) for each group. * *p* < 0.01 as compared to control; **_Ɨ_** *p* < 0.05 as compared to CLP.

**Figure 2 antioxidants-10-01676-f002:**
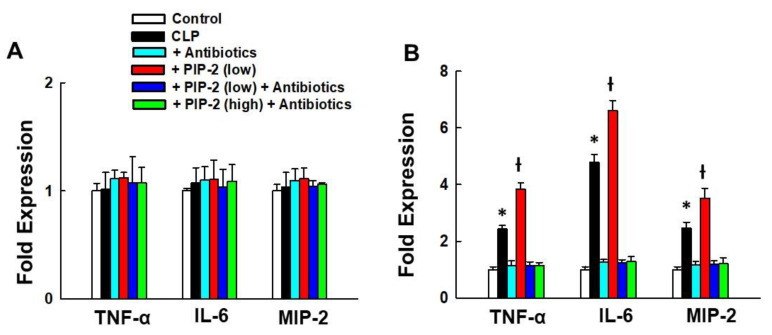
Effect of PIP-2 and/or antibiotics on cytokine expression in lung. Expression of cytokines is expressed as fold change (control = 1) in lung homogenate at 6 h or 24 h after generation of CLP. PIP-2 and /or antibiotics were administered at zero time **(A)** or at 6 h after surgery **(B)**; lungs were harvested at 6 h (A) or 24 h (B) after surgery. Control was normal mice. PIP-2 low, 2 µg/g body weight: PIP-2 high, 10 µg/g body weight. A. mean + range for *n* = 2 for 6 h study; B. mean + SD for *n* = 4 for 24 h study. * *p*< 0.05 vs. all other conditions at 24 h, **ƚ** *p* < 0.05 vs. all other conditions at 24h.

**Table 1 antioxidants-10-01676-t001:** Effect of PIP-2 on aiPLA2 activity of mouse organs.

PLA_2_ Activity nmol/h/mg Protein								
Mouse Organs	Lung		Kidney		Heart		Liver	
	2 h	24 h	2 h	24 h	2 h	24 h	2 h	24 h
Liposomes	8.21 ± 0.28	ND	6.61 ± 0.27	ND	4.75 ± 0.39	ND	5.32 ± 0.33	ND
Liposomes + PIP-2 (2 µg)/g	1.49 ± 0.06	1.51 ± 0.09	1.25 ± 0.06	1.28 ± 0.04	0.95 ± 0.02	0.96 ± 0.06	1.26 ± 0.02	1.27 ± 0.09
Liposomes + PIP-2 (20 µg)/g	1.44 ± 0.06	ND	1.09 ± 0.02	ND	0.85 ± 0.03	ND	1.22 ± 0.04	ND

Mice were injected IV with liposomes ± PIP-2 (2 µg/g or 20 µg/g). After 2 h or 24 h, lungs were cleared of blood, lungs and organs were harvested, and PLA_2_ activity was measured. Results are mean ± range for *n* = 2 for 2 h study and mean ± SD for *n* = 3 for liposomes alone (control) and 24 h study. ND = not determined.

**Table 2 antioxidants-10-01676-t002:** Mortality within 24 h after CLP.

	Number of Mice	Mice Dead by 24 h	% Mortality
CLP	15	1	7
CLP + Antibiotics	13	1	8
CLP + PIP-2 (2 µg/g)	10	0	0
CLP + PIP – 2 (2µ/g) + Antibiotics	11	1	9
CLP + PIP – 2 (10 µg/g)	6	6	100
CLP + PIP – 2 (10 µg/g) + Antibiotics	5	0	0

PIP-2 and /or antibiotics were administered at 6 h after surgery. Mouse survival was observed 18 h later (total time after surgery 24 h.

**Table 3 antioxidants-10-01676-t003:** Bacterial colony formation at 6 h and 24 h after CLP.

	Colonies /mL of Lung Homogenate X 10^−3^		Colonies /mL of Peritoneal Fluid X 10^−3^	
	6 h	24 h	6 h	24 h
Control	0.32 ± 0.10		0.38 ± 0.80	
CLP	0.06 ± 0.05	68.4 ± 21.6 *	10.5 ± 0.2	1522 ± 440 *
CLP + PIP-2 (2 µg/g)	0.09 ± 0.04	133 ± 16.5 ^ƚ^	15.8 ± 0.16	3894 ± 536 ^ƚ^
CLP + Antibiotics	0	0.45 ± 0.14	0	0.30 ± 0.14
CLP + PIP – 2 (2 µg/g) + Antibiotics	0	0.42 ± 0.16	0	0.32 ± 0.12
CLP + PIP – 2 (10 µg/g) + Antibiotics	0	0.37 ± 0.09	0	0.34 + 0.11

PIP-2 and/or antibiotics were administered at zero time for the 6 h study and at 6 h for the 24 h study. At 6 or 18 h after treatments, lung and peritoneal fluid were collected, and peritoneal fluid and homogenized lung were plated on MacConkey medium. Bacterial colonies were counted 24 h later. Results are mean ± range for *N* = 2 for the 6 h study and mean ± SD (*n* = 4) for the 24 h study. * *p*< 0.05 vs. all other conditions at 24 h; ^ƚ^
*p* < 0.05 vs. all other conditions at 24 h.

**Table 4 antioxidants-10-01676-t004:** Indices of lung MPO activity and lung *W/D* at 6 h and 24 h after CLP.

	MPO Activity nmol/mg Protein		Wet/Dry Weight	
	6 h	24 h	6 h	24 h
Control	4.94 ± 3.8		4.90 ± 0.14	
CLP	14.02 ± 7.3	68.7 ±9.8 *	5.13 ± 0.26	7.32 ± 0.51 *
CLP + PIP-2 (2 mg/g)	0.66 ± 0.24	23.3 ± 6.6 * ^ƚ^	5.07 ± 0.23	6.24 ± 0.42 * ^ƚ^
CLP + Antibiotics	0.29 ± 0.14	8.95 ± 4.1 ^ƚ^	5.00 ± 0.22	7.52 ± 0.16 *
CLP + PIP - 2(2 µg/g) + Antibiotics	0.04 ± 0.15	2.40 ± 6.5 ^ƚ^	5.01 ± 0.18	6.27 ± 0.47 * ^ƚ^
CLP + PIP - 2(10 µg/g) + Antibiotics	0.26 ± 0.18	2.94 ± 1.6 ^ƚ^	5.02 ± 0.14	6.23 ± 0.21 * ^ƚ^

Results are mean ± range for *n* = 2 for the 6 h study (cohort 1) and mean ± SD for *n* = 4 for the 24 h study (cohort 2). * *p* < 0.05 vs. control; ^ƚ^
*p* < 0.05 vs. CLP at 24 h. Control is normal untreated mice.

**Table 5 antioxidants-10-01676-t005:** Oxidative injury measured at 6 h and 24 h after CLP.PIP-2 and /or antibiotics were administered at zero time (6 h) or at 6 h after surgery (24 h).

	TBARS pmol/mg Prot.		8-Isoprostanes pmol/mg Prot.		Protein Carbonyls nmol/mg Prot.	
	6 h	24 h	6 h	24 h	6 h	24 h
Control	69.3 ± 2.6		32.3 ± 1.6		5.25 ± 0.40	
CLP	73.4 ± 5.9	243 ± 12 *	32.9 ± 1.4	121 ± 5.8 * ^ƚ^	5.26 ± 0.02	16.1 ± 1.26 *
CLP + PIP-2 (2 µg/g)	72.8 ± 2.2	105 ± 5.5 * ^ƚ^	31.8 ± 4.8	59.6 ± 5.3 * ^ƚ^	5.31 ± 0.74	7.67 ± 0.23 * ^ƚ^
CLP + Antibiotics	71.9 ± 2.8	231 ± 20 *	32.1 ± 1.0	111 ± 9.4 *	5.08 ± 0.33	15.8 ± 0.89 *
CLP + PIP – 2 (2 µg/g) Antibiotics	74.7 ± 5.6	108 ± 13 * ^ƚ^	33.7 ± 2.0	53.4 ± 6.3 * ^ƚ^	5.19 ± 0.33	7.74 ± 0.28 * ^ƚ^
CLP + PIP – 2 (10 µg/g) Antibiotics	74 .5 ± 7.2	111 ± 2.6 * ^ƚ^	32.7 ± 1.8	56.6 ± 6.5 * ^ƚ^	4.96 ± 0.33	7.60 ± 0.15 * ^ƚ^

Control is normal untreated mice. Results are mean ± range for *n* = 2 for 6 h study and mean ± SD for *n* = 4 for 24 h study. * *p* < 0.05 vs. control; ^ƚ^
*p* < 0.05 vs. CLP alone at 24 h.

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
