# Peer review of "Inhibition of Peroxiredoxin 6 PLA2 Activity Decreases Oxidative Stress and the Severity of Acute Lung Injury in the Mouse Cecal Ligation and Puncture Model"

_antioxidants, 2021, doi:10.3390/antiox10111676_

Round 1

Reviewer 1 Report

In previous studies the authors identified a 9 aa peptide derived from surfactant protein-A which inhibits aiPLA2 activity. The peptide is called peroxiredoxin 6-inhibitory peptide-2 (PIP-2). PIP-2 appears to be specific for inhibition of aiPLA2 activity. In further studies they showed that PIP-2 protects against lung injury in both LPS and ventilator-induced lung injury models of ALI in mice.

In their recent study here they extended the use of PIP-2 in a mouse model of sepsis to study effectivity of the peptide in a non-sterile model of systemic infection/inflammation. Although PIP-2 was able to reduce the inflammation mice died from sepsis when they were not treated in parallel with antibiotics. The authors conclude that suppression of the immune response by PIP-2 interferes with killing of bacteria.

One important paper published in the blue journal in 2016 by Amatullah et al (PubMed: 27735193) showed that the endogenous regulatory protein DJ-1 inhibits the production of ROS, therefore interfering with bacterial clearance in the CLP model. By knocking out DJ-1 the authors showed enhanced bacterial killing and an overall improved survival and organ function despite high levels of ROS. How do these data fit to the study that is presented here?

In the conclusion the authors state: “We propose that the addition of PIP-2 to the current regimen has the potential to inhibit ROS-mediated lung injury and to lower the currently unacceptably high mortality rate for ALI.” I do not think that this is a valid conclusion since the authors showed in their paper that inhibition of ROS leads to enhanced dying of mice. Only when the bacterial growth is adequately controlled with antibiotics the treatment can have some benefit. However, what would happen if bacterial growth could not be controlled by antibiotics in the septic patient e.g. due to resistant strains? Would PIP-2 treatment lead to more dying patient?

In table 1 controls are missing: 1) empty liposomes versus untreated, to determine whether liposomes have an effect by themselves  2) PIP2 non-encapsulated versus encapsulated to determine whether liposome encapsulation is advantageous compared to the free molecule.

Chapter 3.2 which concentration was used? 2µg/g as indicated in the text or 2mg/g as stated in the table?? Was the peptide encapsulated or not??

Table 3 why is no data shown for the group CLP 10µg/g without antibiotics? Although the mice had to be euthanized before the terminal endpoint of the experiment was reached, the authors could have collected the lungs and did all the different measurements…

Figure 2 is not well presented. The resolution of the figure is bad and must be improved. The different symbols indicating statistical differences are not easy to identify. There are many unneeded paragraph marks in the figure. Overall I would suggest to use colored columns. Moreover, it is also worth to think about transferring some of the tables also into diagrams to make the paper more appealing for the reader.

Imipenem, which was used for antibiotic treatment of mice, is also thought to have an immunomodulatory effect e.g. doi: 10.1093/jac/44.4.561. The authors did not account for this. Therefore the authors should at least discuss its possible impact. 

Author Response

REVIEWER 1:

In previous studies the authors identified a 9 aa peptide derived from surfactant protein-A which inhibits aiPLA2 activity. The peptide is called peroxiredoxin 6-inhibitory peptide-2 (PIP-2). PIP-2 appears to be specific for inhibition of aiPLA2 activity. In further studies they showed that PIP-2 protects against lung injury in both LPS and ventilator-induced lung injury models of ALI in mice.

In their recent study here, they extended the use of PIP-2 in a mouse model of sepsis to study effectivity of the peptide in a non-sterile model of systemic infection/inflammation. Although PIP-2 was able to reduce the inflammation mice died from sepsis when they were not treated in parallel with antibiotics. The authors conclude that suppression of the immune response by PIP-2 interferes with killing of bacteria. One important paper published in the blue journal in 2016 by Amatullah et al (PubMed: 27735193) showed that the endogenous regulatory protein DJ-1 inhibits the production of ROS, therefore interfering with bacterial clearance in the CLP model. By knocking out DJ-1 the authors showed enhanced bacterial killing and an overall improved survival and organ function despite high levels of ROS. How do these data fit to the study that is presented here?

Our response: DJ-1 is an oncogenic protein that has some antioxidant properties based on several cysteines. We don’t see any direct connection to PIP-2 or NOX2 activation.

In the conclusion the authors state: “We propose that the addition of PIP-2 to the current regimen has the potential to inhibit ROS-mediated lung injury and to lower the currently unacceptably high mortality rate for ALI.” I do not think that this is a valid conclusion since the authors showed in their paper that inhibition of ROS leads to enhanced dying of mice.

Our response: We have changed this statement to “We propose that the addition of PIP-2 to an effective antibiotic regimen has the potential to inhibit ROS-mediated lung injury and to lower the currently unacceptably high mortality rate for ALI.”

Only when the bacterial growth is adequately controlled with antibiotics the treatment can have some benefit. However, what would happen if bacterial growth could not be controlled by antibiotics in the septic patient e.g. due to resistant strains? Would PIP-2 treatment lead to more dying patient?

Our response: Very likely. We would recommend to NOT use PIP-2 under that circumstance. It is common with the instructions for use of drugs to admonish “do not use if…”

In table 1 controls are missing: 1) empty liposomes versus untreated, to determine whether liposomes have an effect by themselves

Our response: There is no reason to suspect that liposomes serve an important antibacterial function and our previous studies have shown no antioxidant effect of liposomes alone.

2) PIP2 non-encapsulated versus encapsulated to determine whether liposome encapsulation is advantageous compared to the free molecule.

Our response: We have shown previously that liposomes are required for PIP-2 to function intracellularly.

Chapter 3.2 which concentration was used? 2µg/g as indicated in the text or 2mg/g as stated in the table

Our response: 2µg/g Table value is corrected

Was the peptide encapsulated or not??

Our response. Yes. The peptide is not effective for use in cells and intact animals unless encapsulated.

Table 3 why is no data shown for the group CLP 10µg/g without antibiotics? Although the mice had to be euthanized before the terminal endpoint of the experiment was reached, the authors could have collected the lungs and did all the different measurements.

Our response: The animals were NOT euthanized. They were dead when the tech arrived in the AM. We could arrange the timing to get animals before they died but I doubt that our IACUC would approve, and we would be reluctant to ask. We believe that the data show an increase in bacterial load with the high dose of PIP-2. The exact increase in load at some intermediary time point would not help us to understand the reason for the mortality.

Figure 2 is not well presented. The resolution of the figure is bad and must be improved. The different symbols indicating statistical differences are not easy to identify. There are many unneeded paragraph marks in the figure.

Our response: Figure 2 has been improved.

Overall I would suggest to use colored columns. Moreover, it is also worth to think about transferring some of the tables also into diagrams to make the paper more appealing for the reader.

Our response: Yes—thanks for the suggestion. We are used to publishing in journals where Figs unnecessarily in color is discouraged, but we agree that it fits well here.

Imipenem, which was used for antibiotic treatment of mice, is also thought to have an immunomodulatory effect e.g. doi: 10.1093/jac/44.4.561. The authors did not account for this. Therefore, the authors should at least discuss its possible impact.

Our response: We have added an explanation for this in the Discussion section. Specifically we stated “It should be noted that the antibiotic imipenem and possibly other antibiotics when administered long term have some immunomodulatory effect [55]; so the decreased cytokine release with antibiotics in the present study may possibly be related in part to the immune activity of imipenem.”

Reviewer 2 Report

This study showed that the administration of PIP-2 might protect against lung injury in a murine model of sepsis-induced ALI when antibiotics were concurrently used. Whereas PIP-2 treatment alone deteriorated lung injury and worsened the outcome due to decreased lung bacterial clearance. This impaired bactericidal activity was attributable to NOX2 inhibition by reducing Peroxiredoxin 6 PLA2 activity. These findings are interesting but there are some limitations in this study:

  1. In this study, there was no direct evidence to prove that PIP-2 inhibits the bactericidal activity of PMN and thereby limit clearance of bacteria from the lung. It is important to assess the function or phagocytic activity of PMN with and without the treatment of PIP-2.
  2. This study showed that high dose PIP-2 was lethal by increasing bacterial infection in the absence of antibiotics, but low dose PIP-2 was not. However, low dose PIP-2 was equally effective in the inhibition of NOX2 compared to the high dose. Is it possible that high dose of PIP-2 can not only inhibit phagocytic activity of PMN but also somehow induce cell apoptosis? However, no data were found in this study. In addition, bacterial cultures were not done for the high dose PIP-2 lungs due to all these mice died during the 24 h after CLP. Nevertheless, these data may help explain the high mortality rate.
  3. As mentioned in the manuscript, inhibition of aiPLA2 activity at 24 h after low dose PIP-2 was similar to the 2 h value. However, the inhibitory effect of aiPLA2 activity at 24 h in high dose (20 μg) was not measured. Therefore, it is not clear whether the actual efficacy of high dose PIP-2 on inhibition of NOX2 is same as the pattern of low dose PIP-2.
  4. In the cohort 1 with ALI preventive strategy, the observation period was only 6h instead of 24h. Therefore, the preventive effect is unable to observe as the results of W/D ratio, lung cytokines and lung oxidative stress. Also, it is unlikely to compare 2 cohorts to identify which strategy has better outcomes.
  5. The statement “Thus, any NOX2 inhibitor should have essentially the same effect on bacterial load as shown for PIP-2. Likewise, any effective ROS scavenger also is likely to decrease the clearance of bacteria” should have evidence or references.
  6. Page 5, line 216, in the sentence “administration of aiPLA2 was similar to the ROS production in lungs from NOX2 null mice.” Should “aiPLA2” be “MJ33?”

Author Response

REVIEWER 2:

This study showed that the administration of PIP-2 might protect against lung injury in a murine model of sepsis-induced ALI when antibiotics were concurrently used. Whereas PIP-2 treatment alone deteriorated lung injury and worsened the outcome due to decreased lung bacterial clearance. This impaired bactericidal activity was attributable to NOX2 inhibition by reducing Peroxiredoxin 6 PLA2 activity. These findings are interesting but there are some limitations in this study

  1. In this study, there was no direct evidence to prove that PIP-2 inhibits the bactericidal activity of PMN and thereby limit clearance of bacteria from the lung.

Our response: We have shown previously that PIP-2 inhibits ROS production by the lung. Numerous previous studies have shown that NOX2 kills bacteria via ROS production. When we treat the mice with PIP-2, the bacterial count in the lungs and peritoneal fluid increases significantly. Ergo….

It is important to assess the function or phagocytic activity of PMN with and without the treatment of PIP-2.

Our response: Our lab is not in a position to study phagocytic activity.

  1. This study showed that high dose PIP-2 was lethal by increasing bacterial infection in the absence of antibiotics, but low dose PIP-2 was not. However, low dose PIP-2 was equally effective in the inhibition of NOX2 compared to the high dose. Is it possible that high dose of PIP-2 can not only inhibit phagocytic activity of PMN but also somehow induce cell apoptosis?

Our response: We suppose that there are many possibilities as to why high dose PIP-2 was lethal. Apoptosis is one possibility although we see no evidence to support it. Clearly, we would like to know the “reason” for the mortality, but that will take an intense effort and probably some time to accomplish. Even if we found apoptosis (for which we see no evidence), that would get us no closer to a true understanding since we would not know why apoptosis occurred.

However, no data were found in this study. In addition, bacterial cultures were not done for the high dose PIP-2 lungs due to all these mice died during the 24 h after CLP. Nevertheless, these data may help explain the high mortality rate.

Our response: We believe that there is ample evidence that the bacterial count increased with high PIP-2. Whether it increased 2-fold or 10-fold is not really important to the conclusions. As explained in response to reviewer 1, we are reluctant to pursue those studies based on humane considerations

  1. As mentioned in the manuscript, inhibition of aiPLA2 activity at 24 h after low dose PIP-2 was similar to the 2 h value. However, the inhibitory effect of aiPLA2 activity at 24 h in high dose (20 μg) was not measured. Therefore, it is not clear whether the actual efficacy of high dose PIP-2 on inhibition of NOX2 is same as the pattern of low dose PIP-2.

Our response: With in vitro testing, the effects are the same with high vs low dose of PIP-2.

  1. In the cohort 1 with ALI preventive strategy, the observation period was only 6h instead of 24h. Therefore, the preventive effect is unable to observe as the results of W/D ratio, lung cytokines and lung oxidative stress. Also, it is unlikely to compare 2 cohorts to identify which strategy has better outcomes.

Our response: We do not see any reason to compare the 2 strategies. Prevention could be used with a patient at high risk for developing ALI. If a patient has ALI, prevention is no longer an option. The determination as to whether use for prevention is acceptable will depend on a future in depth study of safety and side effects of the agent.

  1. The statement “Thus, any NOX2 inhibitor should have essentially the same effect on bacterial load as shown for PIP-2. Likewise, any effective ROS scavenger also is likely to decrease the clearance of bacteria” should have evidence or references.

Our response: We don’t think that there are comprehensive references to corroborate this. We have reworded the sentence to rely on logic rather than experiment.

  1. Page 5, line 216, in the sentence “administration of aiPLA2 was similar to the ROS production in lungs from NOX2 null mice.” Should “aiPLA2” be “MJ33?”

Our response: Yes, you are correct. Thank you.

Round 2

Reviewer 1 Report

Thank you for taking my suggestions into account.

Author Response

We thank this reviewer for his/her critiques which has significantly improved the manuscript.

Reviewer 2 Report

This study suggests that PIP-2 inhibits the bactericidal efficacy of PMN resulting in increased lung infection. However, in this study, we only found that the result of MPO activity measurement could partially support this statement. As we have known that not only PMNs but also other immune cells (such as macrophagy) can be responsible for bacterial clearance, how the authors can ensure that PIP-2 mainly acts on PMNs rather than other immune cells.

Line 274, the sentence “PIP-2 treatment (low dose) resulted in a significant decrease in the W/D at 24 h compared with PIP-2 alone” seems not correct. Should “compared with PIP-2 alone” be “compared with CLP alone?”

Line 374, the statement “studies with both the preventative and treatment protocols showed that PIP-2 had a significant beneficial effect on the measured indices of lung injury.” However, in cohort 1 studying preventative effect, the results of W/D ratio (table 4), cytokine expression (figure 2) and oxidative injury (table 5) all showed not significant. It is suggested to change the statement and have some discussion.

Author Response

REVIEWER 2:

This study suggests that PIP-2 inhibits the bactericidal efficacy of PMN resulting in increased lung infection. However, in this study, we only found that the result of MPO activity measurement could partially support this statement. As we have known that not only PMNs but also other immune cells (such as macrophagy) can be responsible for bacterial clearance, how the authors can ensure that PIP-2 mainly acts on PMNs rather than other immune cells.

Our response: We added a statement in the Discussion section to address this. Specifically we state (line 335) “Thus, PIP-2 inhibits the bactericidal efficacy of PMN as well as that of macrophages and possibly other cells associated with immune function resulting, by definition, in increased lung infection”.

Line 274, the sentence “PIP-2 treatment (low dose) resulted in a significant decrease in the W/D at 24 h compared with PIP-2 alone” seems not correct. Should “compared with PIP-2 alone” be “compared with CLP alone?”

Our response: Thank you. The error has been corrected.

Line 374, the statement “studies with both the preventative and treatment protocols showed that PIP-2 had a significant beneficial effect on the measured indices of lung injury.” However, in cohort 1 studying preventative effect, the results of W/D ratio (table 4), cytokine expression (figure 2) and oxidative injury (table 5) all showed not significant. It is suggested to change the statement and have some discussion.

Our response: We expanded on the discussion to address this. Specifically we state “Lung injury at 6 h after CLP was relatively slight and was effectively treated with antibiotics. Thus, a beneficial effect of PIP-2 was not seen at 6 h (the prevention protocol) in these minimally injured lungs. However, studies at 24 h after CLP (the treatment protocol) showed that PIP-2 had a significant beneficial effect on the measured indices of lung injury compared to treatment with antibiotics alone.”
